# DepthLM: Metric Depth From Vision Language Models

**Zhipeng Cai**[1*]**, Ching-Feng Yeh**[1]**, Xu Hu**[1]**, Zhuang Liu**[2]**, Gregory P. Meyer**[1]**, Xinjie Lei**[1]**,
Changsheng Zhao**[1]**, Shang-Wen Li**[1]**, Vikas Chandra**[1]**, Yangyang Shi**[1]
[1]Meta, [2]Princeton University
[*] Project lead & corresponding author (czptc2h@gmail.com)

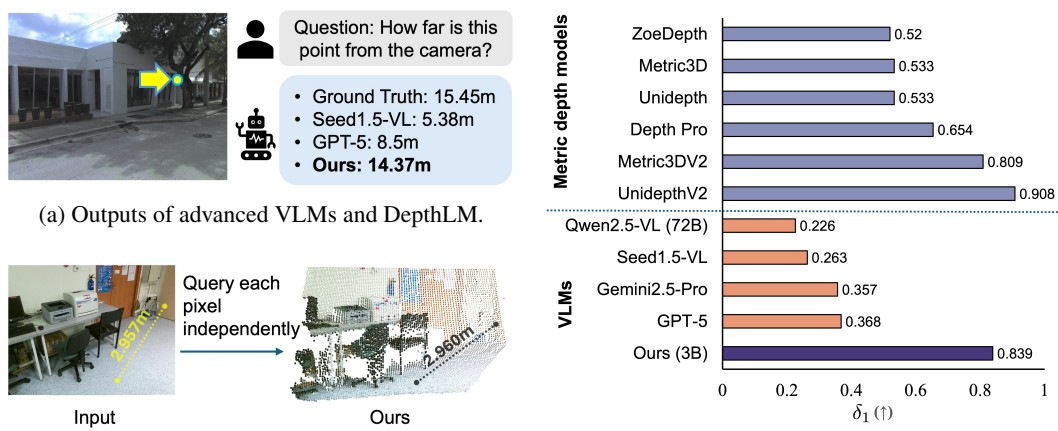

(a) Outputs of advanced VLMs and DepthLM.

(b) Point clouds generated by DepthLM.

(c) Accuracy $\delta_1(\uparrow)$ over 4 datasets (Nuscenes, ETH3D, sunRGBD, ibims1).

Figure 1: **We propose DepthLM, a simple and effective method that turns VLMs into strong pixel-level metric depth estimators.** The latest VLMs, including GPT-5, still struggle in understanding 3D from 2D inputs. Our model is the *first* VLM that has comparable accuracy to advanced metric depth models, and can generate point clouds with accurate metric scales.

## Abstract

Vision language models (VLMs) can flexibly address various vision tasks through text interactions. Although successful in semantic understanding, state-of-the-art VLMs including GPT-5 still struggle in understanding 3D from 2D inputs. On the other hand, expert pure vision models achieve super-human accuracy in metric depth estimation, a key 3D understanding task. However, they require task-specific architectures and losses. Such difference motivates us to ask: Can VLMs reach expert-level accuracy without architecture or loss change? We take per-pixel metric depth estimation as the representative task and show that the answer is *yes*! Surprisingly, comprehensive analysis shows that text-based supervised-finetuning with sparse labels is sufficient for VLMs to unlock strong 3D understanding, *no* dense prediction head or complex regression/regularization loss is needed. The bottleneck lies in *pixel reference* and *cross-dataset camera ambiguity*, which we address through visual prompting and intrinsic-conditioned augmentation. With much smaller models, our method *DepthLM* surpasses the accuracy of most advanced VLMs by over 2x, making VLMs *for the first time* comparable with pure vision models. The simplicity of DepthLM also enables a single VLM to cover various 3D tasks beyond metric depth. Code and model are available at
https://github.com/facebookresearch/DepthLM_Official.

# 1 INTRODUCTION

Understanding 3D from 2D inputs lies at the core of many applications, such as self-driving and robotics. Pure vision models (Hu et al., 2024; Bochkovskii et al., 2024) achieve super-human accuracy through task-specific architectures and complex training losses, which often require drastically different design for different tasks. Vision language models (VLMs) (Liu et al., 2023) connect visual inputs with language models, making it possible for a unified model to flexibly handle diverse vision tasks through language interactions. However, Fig. 1 shows that even the most advanced VLMs still struggle with basic 3D understanding, under-performing pure vision models by a large margin.

In this work, we use the classic 3D understanding task, *pixel-level metric depth estimation*, as the representative and conduct comprehensive analysis on different aspects of VLMs. Surprisingly, we show that the poor 3D understanding of VLMs is not due to the lack of dense prediction head or complex regression/regularization losses. We propose *DepthLM*, a simple yet effective method that can turn VLMs into strong metric depth estimators without loss or architecture change.

At the core of DepthLM are 1) *visual prompting*, where we render markers on images rather than putting coordinates in text prompts, so that VLMs can accurately understand pixel locations, 2) *intrinsic-conditioned augmentation*, where we unify the focal length of different images to resolve camera ambiguity and enable zero-shot generalization, 3) *sparsely labeled images with text-based supervised fine-tuning (SFT)* (Wei et al., 2021), where we can use 1 labeled pixel per training image to enable strong 3D understanding. We also investigate reinforcement learning (RL) (Kaelbling et al., 1996) for model training. Interestingly, both SFT and RL can learn 3D understanding, while SFT is more efficient. We curate public datasets into a VLM benchmark suite, *DepthLMBench*, to train VLMs and directly compare with pure vision models in 3D understanding.

With a 3B model, DepthLM outperforms most advanced and much larger VLMs including GPT-5 (Singh et al., 2025), achieving over 2x accuracy ($\delta_1$ (Ranftl et al., 2020)) improvement across 4 indoor and outdoor datasets (Fig. 1c). DepthLM also outperforms DepthPro (Bochkovskii et al., 2024) and Metric3Dv2 (Hu et al., 2024), demonstrating for the *first* time that VLMs can match the accuracy of advanced pure vision models, and generate a high quality point cloud of an image with an accurate metric scale (Fig. 1b). Importantly, all these results are achieved with standard VLMs and text interactions, *no* dense prediction head or extra module is needed. The simplicity of DepthLM also makes it more flexible and scalable than expert pure vision models. With the same framework, DepthLM naturally allows us to train a unified VLM, achieving high accuracy on diverse and complex 3D tasks involving reasoning, multi-point and multi-image understanding.

# 2 RELATED WORK

**Metric Depth Estimation.** Metric depth estimation is a classic 3D understanding task that predicts the depth of a pixel in meters. ZoeDepth (Bhat et al., 2023) and Metric3D series (Yin et al., 2023; Hu et al., 2024) pioneered the efforts of zero-shot metric depth estimation. ZoeDepth trained two prediction heads for indoor and outdoor scenes respectively, and used a router to automatically decide which head to use given the input. Metric3D leveraged the camera intrinsics information to normalize either the input images or the output labels so that the model could effectively learn a unified metric scale of the world even when mixing with data from different types of cameras. To remove the need of camera intrinsics, recent works (Bochkovskii et al., 2024; Piccinelli et al., 2024; 2025; Wang et al., 2025; Zeng et al., 2024) designed specific architectures or leverage language priors to predict the intrinsics of the input directly. This work shows that handling camera ambiguity is also essential in VLMs. Through comprehensive analysis, we find that augmenting inputs is more effective for VLMs than other approaches.

**VLMs for 3D understanding.** SpatialVLM (Chen et al., 2024) pioneered the work of 3D understanding with VLMs. It turned the output of pure vision models into text prompts for training. SpatialRGPT (Cheng et al., 2024) extended this idea where besides using templates to generate the prompts, it leveraged text-based LLMs to generate complex reasoning questions. However, knowledge distillation from a complex pipeline with various pure vision models will accumulate errors in training data. Focusing on object-level tasks also makes these VLMs hard to compare with pure vision models, making it unclear on how far existing VLMs are from advanced pure vision models. SpatialBot (Cai et al., 2024) leverages the depth map derived from pure vision models (Bhat et al.,

2023) to enable pixel-level 3D understanding. Such setting not only requires extra modules in the architecture, but also limits the tasks it can handle. Without involving more expert vision models, it struggles in more complex multi-image tasks such as camera pose estimation. Recent works (Xu et al., 2025; Guo et al., 2025) also studied pixel-level metric depth estimation with VLMs. However, they did not release models or investigate the cause of the performance gap between VLMs and pure vision models. As a result, their accuracy was still distant from pure vision models.

## 3 DEPTHLM

To understand why VLMs are behind pure vision models on 3D understanding, we first conduct comprehensive analysis on major VLM components except the architecture that we do not change. They include prompt design (Sec. 3.1), training losses (Sec. 3.2) and mix-data training (Sec. 3.2). Then we propose our final method inspired by the analysis findings (Sec. 3.4).

**DepthLMBench.** We curate a mixture of public datasets widely used in pure vision models into DepthLMBench, a new benchmark suite to enable the training of VLMs for 3D understanding and directly compare with pure vision models during evaluation. For both training and evaluation data, we convert the numerical 3D labels into text with template based approaches in Sec. 3.1.

For training, we mix 7 widely used datasets in pure vision models with indoor and outdoor scenes. For outdoor scenes, we use Argoverse2 (Wilson et al., 2023), Waymo (Mei et al., 2022), and NuScenes (Caesar et al., 2020). For indoor scenes, we use ScanNet++ (Yeshwanth et al., 2023), Taskonomy (Zamir et al., 2018), HM3d (Ramakrishnan et al., 2021) and Matterport3d (Chang et al., 2017). This results in around 16M training images (see Appendix A.2 for details). Pure vision models (Bochkovskii et al., 2024; Piccinelli et al., 2025) often mix > 20 datasets for training, including a large portion of synthetic data. Empirically, low quality and synthetic data do not benefit VLM training without cleaning, hence we only use the high quality datasets above for simplicity.

For evaluation, we mix 8 datasets including both indoor and outdoor scenes non-overlapping with the training data, which allows us to thoroughly evaluate the performance of VLMs on metric depth estimation. There are 3 outdoor datasets: Argoverse2, DDAD (Guizilini et al., 2020), NuScenes; 4 indoor datasets: ScanNet++, sunRGBD (Song et al., 2015), iBims1 (Koch et al., 2018) and NYUv2 (Silberman et al., 2012); and 1 dataset ETH3D (Schops et al., 2017) with both indoor and outdoor scenes. Since existing evaluation datasets with accurate labels are limited, we still include data from Argoverse2, Nuscenes and Scannet++ to ensure scene diversity, and reserve around 10 scenes per dataset that are non-overlapping with the training data for evaluation.

**Analysis Setup.** Unless otherwise stated, we finetune DepthLM on the pre-trained 3B model of (Bai et al., 2025) using text-based supervised fine-tuning (SFT) for analysis. See Sec. 4 for implementation details. We follow the standard in pure vision models (Bochkovskii et al., 2024) and use $\delta_1$ to evaluate the prediction accuracy, i.e., ratio of the outputs that are within $25\%$ difference to the ground-truth (GT). We evaluate on each dataset for 8,192 random samples, which provides stable statistics, i,e, increasing this number introduces negligible difference.

### 3.1 PROMPT DESIGN

We start our analysis from the simplest setup where the training and evaluation data are from the same dataset to remove issues from mix-data training.

**Text or marker-based pixel reference?** VLMs have no pre-defined grid-shaped output domain as pure vision models. How to prompt them to understand fine-grained pixel locations is an important question, since slight error could make the model output depth on a completely different object, leading to severe errors. Existing VLMs (Cai et al., 2024; Guo et al., 2025) often refer to a pixel using its coordinate (X, Y) in text. However, our initial experiments show that VLMs struggle to map text-based coordinates to the pixel locations, even when trained with them. We find that visual prompting is an effective solution to this issue.

For visual prompting, we render markers pointing to the query pixel directly on the input image, and ask: *"How many meters is this point from the camera?"*. See Appendix A.1 for the rendered markers. During training, we set the answer to *"The point is around X meters away from the camera."*, where X is treated as normal text and is rounded to two decimal places, which prevents long

floating point values and is sufficient for typical 3D understanding tasks. Empirically, VLMs are not sensitive to the prompt we use. We do not ask for the principal (forward-backward) axis distance as in pure vision models (Bhat et al., 2023) since euclidean distance is more common in language interactions and the two values can convert to each other using camera intrinsics. Sec. 3.4 shows the prompt that can enable VLMs to predict principal axis distance.

As shown in Fig. 2, text-based pixel reference (similar to (Guo et al., 2025), see Appendix A.1) degrades the model accuracy on both indoor and outdoor data. The gap is especially large (0.15) on indoor scenes (ScanNet++), we conjecture that this is because more objects and occlusions are presented in indoor scenes, leading to more boundary regions. Meanwhile, the accuracy remains similar with different marker shapes, showing the robustness of visual prompting. Interestingly, Sec. 4 shows that even for VLMs trained with text-based coordinates, applying visual prompting still benefits the accuracy. In summary

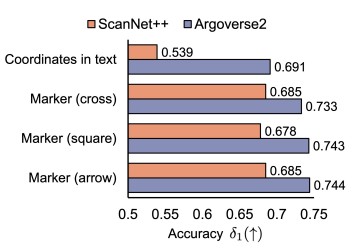

Figure 2: **Pixel reference.**

**Finding 1.** *VLMs understand marker-based pixel reference much better than text-based one.*

## 3.2 LOSS

**SFT or RL?** Given the advancement of reasoning, reinforcement learning (RL) especially GRPO (Shao et al., 2024) has become a popular alternative of SFT. To scale up training, it is important to understand which method is more suitable for 3D understanding. GRPO leverages format-based rewards for LLM training. Since metric depth estimation, as many 3D understanding tasks, returns numbers as the answer, we can use regression-based metrics as the reward. We apply the GRPO style prompt and ask: *"How far is this point from the camera? Output the thinking process in <think> </think> and final answer (the meter number only, without the unit) in <answer> </answer> tags."*. We extract X from templates to compute the reward. Empirically, the model shares similar reasoning traces for all inputs after GRPO training. An example answer is: *"<think> The point is located about X meters from the camera. </think>, <answer> X </answer>"*.

As shown in Fig. 3a, GRPO is not very sensitive to the reward function, we experiment with 1) $\delta_1$, 2) negative $AbsRel$, i.e., the negative value of absolute relative difference (Ranftl et al., 2020), 3) the negative $\mathcal{L}_2$ loss, and 4) the negative $\mathcal{L}_1$ loss. All metrics have reasonable accuracy. Overall, the negative $\mathcal{L}_1$ loss is a simple and effective reward. With this reward, we tuned other GRPO hyper-parameters (see Appendix A.4), and compare GRPO with SFT. As shown in Fig. 3b, given the same number of training samples, SFT and GRPO perform similarly well. However, GRPO requires much heavier per-sample compute than SFT (8-16 times slower in practice). Therefore, we conclude that

**Finding 2.** *Though both SFT and RL can achieve reasonable accuracy, SFT is more efficient to scale up VLM training for 3D understanding.*

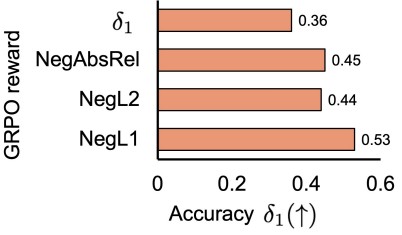

(a) Effect of GRPO rewards (8K training samples).

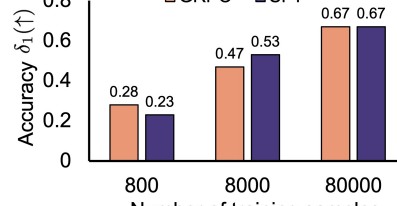

(b) Accuracy vs training data size.

Figure 3: **SFT vs GRPO.** (a) Negative $\mathcal{L}_1$ (NegL1) is the best GRPO reward while most common metrics have reasonable accuracy. (b) While SFT and GRPO achieve similar accuracy given the same train dataset size, SFT has much higher per-sample efficiency. We use Argoverse2 dataset for experiments, see Appendix A.3 for cross dataset evaluation, which shows the same trend.

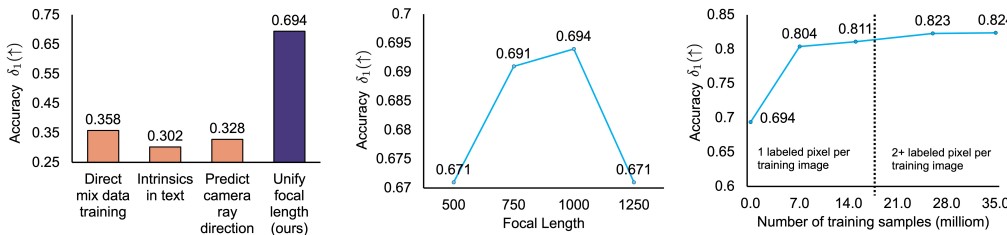

(a) Accuracy of different camera ambiguity handling strategies. (b) Increasing $f_{uni}$ benefits the performance until 1000 pixels. (c) Accuracy with different number of training samples.

Figure 4: **Mix data training analysis.** For (a) and (b), we train on 500K samples on the mixed datasets of DepthLMBench, and report the average accuracy across all evaluation datasets.

### 3.3 MIX-DATA TRAINING

**How to handle camera ambiguity in VLMs?** An important problem in metric depth mix-data training is camera ambiguity, where different images can be captured by different cameras. Similarly looking images from different cameras can have drastically different scale. Directly mix such data for training is sufficient for relative depth (Ranftl et al., 2020). However, it makes the model struggle to learn a generalizable world scale, failing on metric depth estimation (Yin et al., 2023). Different strategies have been proposed to handle camera ambiguity in pure vision models (Bhat et al., 2023; Yin et al., 2023; Bochkovskii et al., 2024). For VLMs, Seed1.5-VL (Guo et al., 2025) put camera intrinsics into text prompts. However, what is the most effective approach for VLMs remains unclear.

To answer this question, we compare four popular approaches: 1) Direct training with images of varied focal lengths, hoping VLMs can implicitly learn to distinguish different cameras. 2) Adding camera intrinsics explicitly into the text prompt as in Seed1.5-VL. 3) Predicting camera intrinsics by letting the model output the camera ray direction before metric depth, similar to (Bochkovskii et al., 2024; Piccinelli et al., 2025). 4) Unifying the focal length through intrinsic-conditioned image augmentation, similar to (Yin et al., 2023) (See Sec. 3.4 for details). See Appendix A.5 for the prompts of 2) and 3). As shown in Fig. 4a, unifying the focal length doubles the accuracy of other approaches. Moreover, adding intrinsics explicitly in text or predicting camera intrinsics do not have higher accuracy than direct mix data training. This result shows that

**Finding 3.** *Without architecture change, VLMs struggle to distinguish different cameras, unifying the focal length by intrinsic-conditioned augmentations can effectively resolve camera ambiguity.*

**Best unified focal length.** In intrinsic-conditioned augmentation, we have to decide the value of the unified focal length. As shown in Fig. 4b, increasing the unified focal length, which increases image sizes, benefits the accuracy until 1,000 pixels. Meanwhile, the accuracy changes <3% across a wide range of values, showing the stability of augmentation-based approach.

**How many labeled pixels we need per image?** Pure vision models are trained on millions of densely labeled images, where each image typically involve tens of thousands of labeled pixels. Each VLM training sample only has sparsely labeled pixels by design. How many labeled pixels do VLMs need to see per image in order to catch up with pure vision models is an interesting and unexplored question. Fig. 4c shows the accuracy of DepthLM with different number of training samples. Before the dashed line (16M images), the model only sees 1 labeled pixel per training image, which already achieves over 0.8 $\delta_1$ and as shown later in Sec. 4, already matches the accuracy of pure vision models. Appendix **??** provides further analysis showing that increasing the number of images is more effective than increasing the number of labeled points for DepthLM. This somewhat surprising result shows that

**Finding 4.** *VLMs can learn 3D understanding from as sparse as 1 labeled pixel per training image. Image diversity is more important than label density for VLM training.*

**Increasing the number of images vs increasing label density.** As mentioned in Finding 4, DepthLM benefits more from having more images than having denser labels. To further demonstrate this point, we conduct an experiment using the Argoverse2 dataset by training the model with

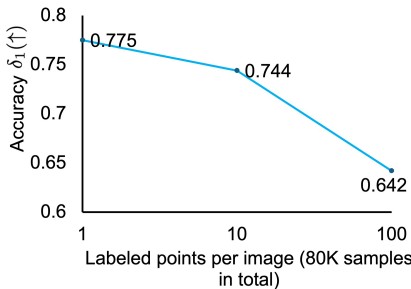

Figure 5: **Increase number of images vs increase label density**. Given the same training dataset size (80K samples), increasing label density while proportionally decreasing the number of images hurts the performance of DepthLM.

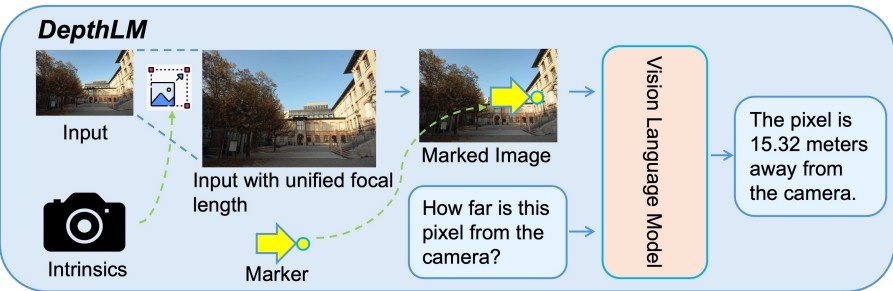

Figure 6: **DepthLM.** DepthLM first augment the input image to have a unified focal length. Then, it renders visual markers on the image for pixel reference and uses text to interact with VLMs directly.

the same number of samples, i.e., question-answer pairs, but vary the label density. Specifically, we train the model on 80K argoverse samples, with 3 label density levels: 1) 1 point per image, i.e., 80K different images with 1 labeled point per image seen during training. 2) 10 points per image, i.e., 8K different images with 10 labeled points per image seen during training. 3) 100 points per image, i.e., 800 different images with 100 labeled points per image seen during training. The evaluation is done on our Argoverse2 evaluation split.

As shown in Fig. 5, increasing the label density, which reduces the number of images, significantly reduces the accuracy. The accuracy difference was larger on $10 \rightarrow 100$ points change than on $1 \rightarrow 10$ points change.

## 3.4 FINAL METHOD

The findings from analysis inspire us to propose the complete method of DepthLM. Fig. 6 shows the method overview.

**Pixel reference.** Based on Finding 1, instead of putting pixel coordinates into the text prompt as in previous VLMs (Cai et al., 2024; Guo et al., 2025), we render visual markers, i.e., small arrows pointing to the pixels, directly on the image to enable accurate pixel location understanding for VLMs. After rendering the visual marker, we can use text prompts to enable various tasks.

**Architecture and training.** We finetune our models from pretrained VLMs without architecture change. Based on Finding 2, we apply standard SFT with the next token prediction paradigm and the cross-entropy loss on the text tokens. As suggested by Finding 4, *no* regression or regularization loss used in pure vision models is needed.

**Resolve camera ambiguity.** Based on Finding 3, we resize the input image before rendering markers to unify its focal length. For simplicity, we assume the input image is undistorted and follows the pinhole camera model (Hartley & Zisserman, 2003). We resize the original input image $\mathcal{I} \in \mathbb{R}^{W \times H \times 3}$ into $\mathcal{I}' \in \mathbb{R}^{W' \times H' \times 3}$ where $W' = \frac{f_{\text{uni}}}{f_x} W$ and $H' = \frac{f_{\text{uni}}}{f_y} H$. $f_x$ and $f_y$ are the focal

length parameters (in pixels) of $\mathcal{I}$, and $f_{uni} = 1000$ is the pre-defined unified focal length. Pure vision models (Yin et al., 2023; Hu et al., 2024) also crop $\mathcal{I}'$ to a fixed size to have unified principle points. As discussed in (Bochkovskii et al., 2024), such strategy makes the model sensitive to the image size, requiring different croping sizes for different evaluation datasets to achieve the reported numbers. In DepthLM, we found that as long as the focal length is unified, we can vary the training image size using random crop so that during evaluation no cropping is needed.

**Beyond metric depth — single VLM for flexible 3D understanding.** We use metric depth estimation to verify that VLMs can achieve comparable accuracy as pure vision models. However, the value of DepthLM is beyond metric depth. Since no task-specific architecture or loss is needed, we can apply the same DepthLM framework to train a unified model on various 3D understanding tasks.

As a concept proof, we expand DepthLM to five other representative tasks. They involve 1) principal axis distance, which estimates the distance in the forward-backward direction rather than the euclidean distance, and is the metric depth that pure vision models actually predict, 2) speed, which estimates the speed needed to reach a point in a limited time, 3) time, which estimates the time needed to reach a point given speed, 4) two point distance, and 5) metric scale camera pose, which estimates the distance that the camera has moved between two images. Fig. 7 visualizes task examples and the outputs of GPT-5 and DepthLM. These tasks cover single image single point questions, reasoning questions that require simultaneous 3D and math understanding, single image multi-point questions, and multi-image questions.

## 4 RESULTS

**Implementation.** To simplify experiments and avoid the need of cropping after unifying the focal length, we use VLMs that can accept images without resizing to a fixed shape. We choose the 3B and 7B models of (Bai et al., 2025) as the default VLM architecture. To demonstrate the general applicability of DepthLM, we also experiment with the 12B model of (Agrawal et al., 2024), which has a significantly different architecture. We train models with PyTorch on 23M-60M samples from DepthLMBench, i.e., 2-4 labeled pixels per image, with 128 H100 GPUs for about 2-4 days depending on the model. We sample equally from each dataset during training except that the weight of Matterport3d is reduced to 0.1 of other datasets since the number of scenes is smaller. During training, we randomly crop the images after normalizing the focal length, with width sampled from 1,000 to 1,400 pixels, and height from 700 to 1,200 pixels. During evaluation, we do not use cropping. See appendix A.6 for detailed hyper-parameters of individual models.

**Main result.** Table 1 and 2 compare DepthLM with existing VLMs and pure vision models respectively. Most VLMs perform not much better than the naive baseline with constant outputs, especially on the indoor data. Even GPT-5 and Gemini-2.5-Pro (Comanici et al., 2025) have below 0.4 $\delta_1$, which is much worse than the worst performing pure vision model as shown in Fig. 1c. Spatial-VLMs like SpaceLLaVA (github contributors, 2024) (third-party implementation of (Chen et al., 2024)) and Spatial-RGPT (Cheng et al., 2024) perform not better than generalist VLMs, though they have been trained for object level 3D understanding. This is not only because they lack

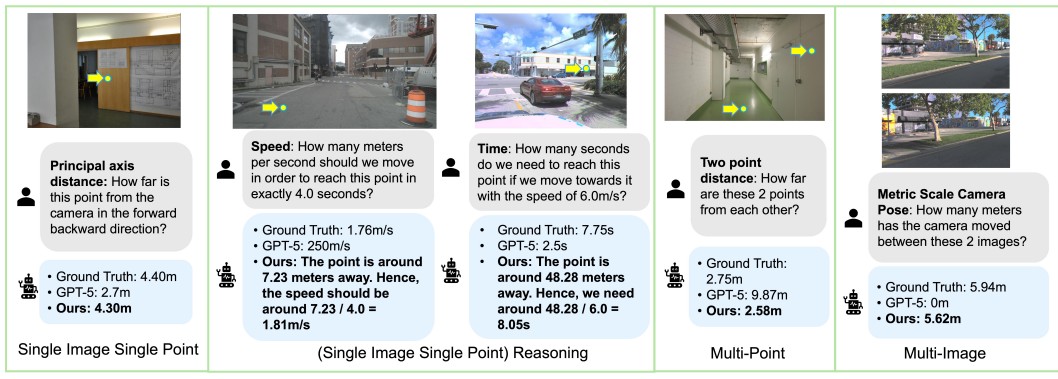

Figure 7: **Scaling DepthLM to more complex 3D tasks.**

Table 1: **VLM result.** We tune the prompt for VLMs that are not trained directly on our task to maximize their performance. State-of-the-art VLMs including GPT-5 have only below 0.4 $\delta_1$. DepthLM, though orders of magnitudes smaller, achieves an over 2x improvement.

| $\delta_1(\uparrow)$ of different methods | Out | | | Out+In | In | | | | Average |
|---|---|---|---|---|---|---|---|---|---|
| | Argoverse2 | DDAD | NuScenes | ETH3D | ScanNet++ | sunRGBD | iBims1 | NYUv2 | |
| *Naive Prediction with Constant Answers* | | | | | | | | | |
| ALWAYS OUTPUT 2.0M | 0.006 | 0.010 | 0.010 | 0.106 | 0.305 | 0.384 | 0.280 | 0.383 | 0.186 |
| *VLMs* | | | | | | | | | |
| QWEN2.5-VL (3B) | 0.133 | 0.083 | 0.090 | 0.087 | 0.120 | 0.134 | 0.080 | 0.128 | 0.106 |
| QWEN2.5-VL (7B) | 0.077 | 0.120 | 0.070 | 0.126 | 0.135 | 0.089 | 0.160 | 0.168 | 0.118 |
| QWEN2.5-VL (72B) | 0.119 | 0.140 | 0.186 | 0.220 | 0.272 | 0.276 | 0.212 | 0.324 | 0.219 |
| MOLMO (7B-D) | 0.200 | 0.132 | 0.200 | 0.126 | 0.244 | 0.299 | 0.200 | 0.225 | 0.203 |
| PIXTRAL (12B) | 0.157 | 0.132 | 0.118 | 0.141 | 0.318 | 0.308 | 0.270 | 0.145 | 0.199 |
| GEMINI-2.5-PRO | 0.280 | 0.252 | 0.365 | 0.328 | 0.380 | 0.270 | 0.466 | 0.394 | 0.342 |
| GPT-O3 | 0.208 | 0.283 | 0.309 | 0.305 | 0.375 | 0.426 | 0.375 | 0.470 | 0.344 |
| GPT-5 | 0.218 | 0.302 | 0.382 | 0.313 | 0.428 | 0.471 | 0.307 | 0.540 | 0.370 |
| *Spatial VLMs* | | | | | | | | | |
| SPACELLAVA (13B) | 0.100 | 0.067 | 0.083 | 0.090 | 0.269 | 0.233 | 0.208 | 0.178 | 0.154 |
| SPATIALRGPT (8B) | 0.055 | 0.046 | 0.100 | 0.220 | 0.346 | 0.369 | 0.240 | 0.265 | 0.205 |
| *VLMs Trained on Metric Depth Estimation* | | | | | | | | | |
| SEED1.5-VL (OFFICIAL SETUP) | 0.009 | 0.012 | 0.013 | 0.219 | 0.495 | 0.321 | 0.459 | 0.412 | 0.243 |
| SEED1.5-VL (OUR PROMPT) | 0.040 | 0.074 | 0.028 | 0.309 | 0.593 | 0.689 | 0.627 | 0.841 | 0.400 |
| OURS (3B) | 0.808 | 0.724 | **0.870** | **0.745** | 0.838 | 0.850 | 0.890 | 0.868 | 0.824 |
| OURS (7B) | **0.833** | **0.747** | 0.865 | 0.718 | **0.850** | **0.859** | **0.920** | **0.915** | **0.838** |
| OURS - PIXTRAL (12B) | 0.734 | 0.670 | 0.819 | 0.653 | 0.834 | 0.786 | 0.870 | 0.799 | 0.771 |

Table 2: **Comparison with pure vision models.** For pure vision models, we use the numbers reported in (Piccinelli et al., 2025), and (Bochkovskii et al., 2024) if some numbers do not exist in (Piccinelli et al., 2025). "-" means no result reported in previous papers. The last column reports the relative accuracy improvement of pure vision models over our model, i.e., $(\delta_1^{\text{CV}} - \delta_1^{\text{Ours}})/\delta_1^{\text{Ours}}$. Our model is the *first* VLM that has comparable accuracy to pure vision models.

| $\delta_1(\uparrow)$ of different methods | Out | | Out+In | In | | |
|---|---|---|---|---|---|---|
| | DDAD | Nuscenes | ETH3D | sunRGBD | ibims1 | vs Ours ($\uparrow$) |
| ZOEDEPTH | 0.272 | 0.283 | 0.350 | 0.867 | 0.580 | -42.8% |
| DEPTHANYTHING | - | 0.354 | 0.093 | 0.850 | 0.714 | -40.3% |
| DEPTHANYTHINGV2 | - | 0.171 | 0.363 | 0.724 | - | -48.5% |
| METRIC3D | - | 0.723 | 0.456 | 0.154 | 0.797 | -36.6% |
| UNIDEPTH | 0.858 | 0.846 | 0.185 | 0.943 | 0.157 | -27.3% |
| DEPTH PRO | 0.299 | 0.566 | 0.397 | 0.831 | 0.823 | -29.1% |
| METRIC3DV2 | - | 0.841 | 0.900 | 0.812 | 0.684 | -3.8% |
| UNIDEPTHV2 | 0.882 | 0.870 | 0.852 | 0.964 | 0.945 | +9.2% |
| OURS (7B) | 0.747 | 0.865 | 0.718 | 0.859 | 0.920 | - |

pixel-level training data, but also because they do not consider camera ambiguity. Seed1.5-VL is the only model that has been trained on the same task as DepthLM, however, its accuracy especially on outdoor data, is still much worse than pure vision models.

Interestingly, instead of using the official setup from its open-source implementation, Seed1.5-VL performs much better when we replace their text-based pixel coordinates with our marker-based pixel reference, i.e., Seed1.5-VL (our prompt), supporting the generalization of our findings. But even with our strategy, It still has only 0.4 $\delta_1$, and has very low accuracy on outdoor data. In contrast, DepthLM has over 0.83 $\delta_1$ which surpasses the accuracy of Depth Pro and Metric3Dv2. This is the *first* time that a VLM can match the accuracy of expert metric depth models on both indoor and outdoor data. More importantly, DepthLM can be flexibly extended to handle various tasks beyond metric depth estimation, with the same architecture and training framework.

In terms of the model size, our 3B model is orders of magnitudes smaller than baseline VLMs like Qwen2.5-VL (72B) and Seed1.5-VL (20B Active MoE Parameters), but still improves their $\delta_1$ by over 2x. If comparing to the base Qwen2.5-VL 3B model, the improvement is about 8x. Our 7B model has slightly higher accuracy than the 3B model, reaching over 0.9 $\delta_1$ on ibims1 and NYUv2. This result shows that larger model sizes help, but they are not necessary for VLMs to have accurate 3D understanding. Meanwhile, our 12B model finetuned from (Agrawal et al., 2024) also has reasonable accuracy, though slightly lower than using the default architecture. This shows the cross-architecture applicability of DepthLM, and the base model performance matters for DepthLM.

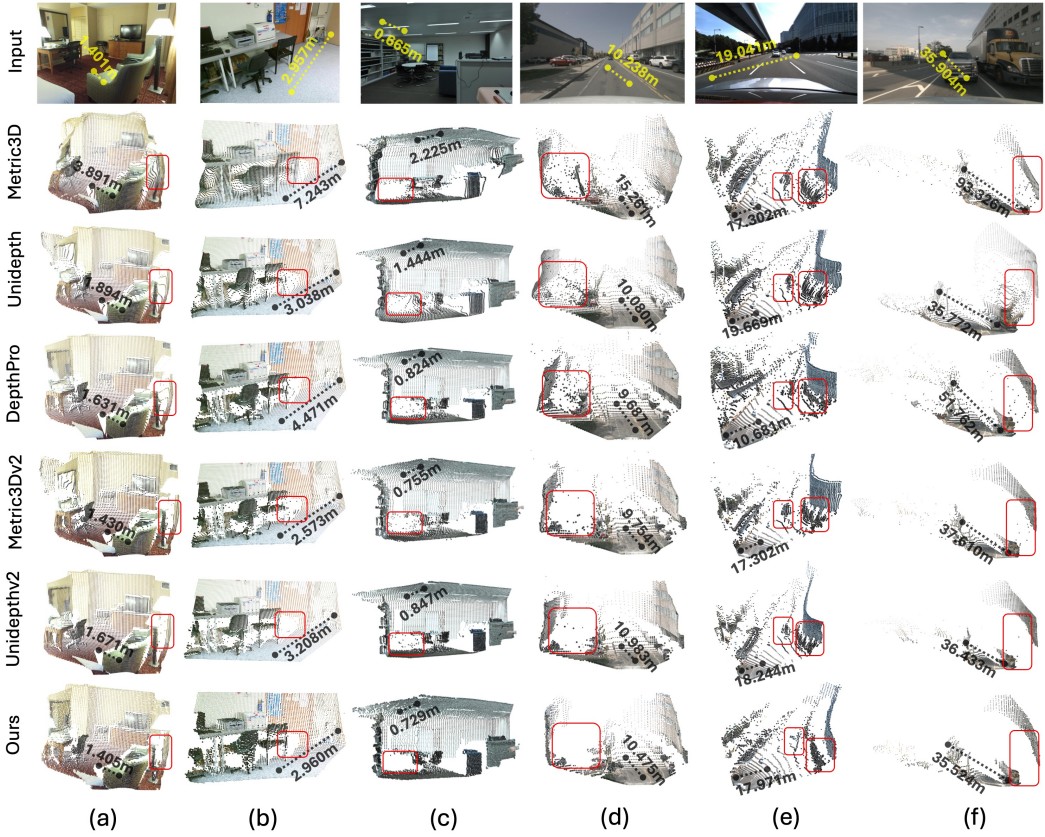

Figure 8: **Visualization.** The scale of DepthLM is close to GT from small indoor scenes to large outdoor scenes. Pure vision models produce smooth points, which is advantageous on non-boundary regions. However, they also have over-smoothing effect on the (marked) boundary regions, leading to flying points between two distinct objects. Interestingly, DepthLM naturally avoids over-smoothing without enforcement during training or any post-processing.

**Visualization.** Interestingly, though only trained for single point prediction, DepthLM can already generate high quality point clouds simply by querying each pixel independently, *without* dense prediction head. Fig. 8 visualizes the point clouds generated by pure vision models and DepthLM (7B). We query on 10K pixels uniformly spanned on each image. The point cloud scale of DepthLM is stable and reasonably close to GT across small indoor scenes and large outdoor scenes. Some pure vision models have severe errors in the scene scale, such as Depth Pro on (b, e, f), Unidepth on (c), and Metric3D on (a, b, c, d, f).

Another interesting observation is that pure vision models and DepthLM have different detail patterns. Specifically, pure vision models generate point clouds with smooth surfaces, however, they also over-smooth the boundary regions as marked in each image, which causes flying points and thin objects merged into other objects (the road lights in (e) and (f)). Interestingly, without *any* post-processing or loss for regularization, DepthLM naturally avoids flying points, showing clear boundaries between different objects, with slightly higher noise in the smooth regions. Such property is beneficial for tasks that require accurate boundary separation, e.g., judging whether a robot can safely move between 2 objects. It also shows that our visual prompting effectively enables accurate pixel location understanding in VLMs, since otherwise the boundaries would not be clear.

**Other Tasks.** To demonstrate the multi-task flexibility of DepthLM, we finetune a unified model on the tasks proposed in Sec. 3.4 together with the metric depth task using our 7b model. The training is done by equally sampling from each task. For simplicity, we only show the average number of each task across multiple datasets. Please refer to Appendix A.7 for results on each dataset. The hyper-parameters are the same as in the main experiments, except that we train for 40M samples.

Table 3: **Multi-task result.** Since not all datasets have pose labels, we train the pose task on Argoverse2 and evaluate on Argoverse2 and Nuscenes.

| $\delta_1(\uparrow)$ on different tasks | Single image single point | | Reasoning | | Multi-point | Multi-image | |
|---|---|---|---|---|---|---|---|
| | Distance | Principal axis distance | Speed | Time | Two point distance | Pose | Average |
| ALWAYS OUTPUT 2.0 | 0.186 | 0.172 | 0.087 | 0.094 | 0.119 | 0.189 | 0.141 |
| QWEN2.5-VL (7B) | 0.118 | 0.085 | 0.136 | 0.087 | 0.066 | 0.048 | 0.09 |
| SPACELLAVA (13B) | 0.154 | 0.163 | 0.116 | 0.122 | 0.157 | 0.047 | 0.127 |
| SPATIALRGPT (8B) | 0.205 | 0.132 | 0.167 | 0.122 | 0.143 | 0.195 | 0.161 |
| SEED1.5-VL (our prompt) | 0.400 | 0.174 | 0.223 | 0.119 | 0.101 | 0.000 | 0.170 |
| Gemini-2.5-Pro | 0.342 | 0.209 | 0.213 | 0.209 | 0.140 | 0.025 | 0.189 |
| GPT-5 | 0.370 | 0.241 | 0.199 | 0.181 | 0.150 | 0.120 | 0.210 |
| OURS (7B) | **0.828** | **0.831** | **0.817** | **0.816** | **0.657** | **0.876** | **0.804** |

As shown in Table 3, without our training, the Qwen2.5-VL (7B) model fails on all tasks, having $<0.1$ $\delta_1$ on average. Advanced proprietary models including GPT-5 also struggle on more complex 3D tasks. As shown in Fig. 7, GPT-5 can return 0m in camera pose estimation when the camera actually moves over 5m. Such catastrophic failure is within expectation since these baselines cannot understand the metric scale even in the basic depth estimation task. DepthLM achieves more than 0.8 $\delta_1$, outperforming the baselines by over 3.8x. Note that the gap between baselines and our model is much larger than in Table 1, showing that VLMs have a larger room to be improved on complex 3D tasks, and the effectiveness of DepthLM. This result also demonstrates the flexibility of DepthLM over expert models, where it can cover diverse tasks with a unified architecture and training framework.

## 5 CONCLUSION

We propose DepthLM, a simple and effective framework that can make VLMs strong pixel-level metric depth estimators. Through comprehensive analysis, we show that VLMs under-perform pure vision models in 3D understanding not because they lack extra modules like dense prediction heads or complex training losses. The key problem lies in pixel reference and camera ambiguity, which we effectively address with the proposed visual prompting and intrinsic-conditioned augmentation approaches. These strategies allow for the first time to use text-based SFT with standard VLMs to match the accuracy of expert pure vision models, and expand the same framework to cover various tasks with a unified model. In terms of limitations, we focus on the simplest and most important design of VLMs. We believe more fine-grained strategies can be investigated in the future to even make VLMs surpass the accuracy of pure vision models. For example, design data filtering pipelines to add more datasets. And training with diverse and complementary tasks to enhance generalization.

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

# A APPENDIX

## A.1 EXAMPLES OF RENDERED MARKERS AND TEXT-BASED PIXEL REFERENCE

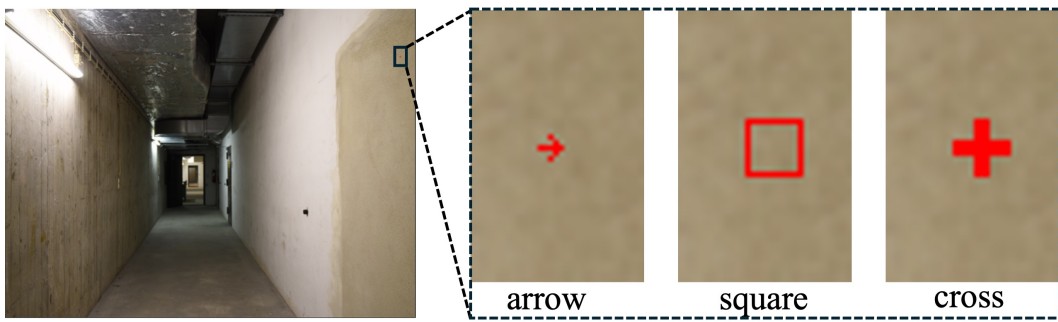

Figure 9: **Actual markers rendered by our method.** We show here 3 different types of markers used in the experiment of Fig. 2.

Fig. 9 shows the actual markers rendered in the experiment of Sec. 3.1. Empirically, a marker of roughly 5 pixels wide is enough for VLMs to recognize it. We also tried different markers and sizes when evaluating baseline VLMs to maximize their accuracy.

For text-based pixel reference experimented in Sec. 3.1, we simply ask: "Given this image of size (width = W, height = H), how far is the pixel at (X, Y) from the camera?"

## A.2 STATISTICS OF DEPTHLMBENCH

Table 4: **Statistics of different training datasets.** We report <images available in the dataset> / <images used for our training>.

| Dataset | Argoverse2 | Waymo | Nuscenes | ScanNet++ | Taskonomy | HM3d | Matterport3d |
|---|---|---|---|---|---|---|---|
| Number of images | 3.5M/1M | 1M/700K | 200K/200K | 10M/1M | 4M/4M | 9M/9M | 190K/190K |

Table 4 shows the number of images for each training dataset in DepthLMBench. Datasets like Argoverse2 and ScanNet++ contain highly similar video frames, which do not help to improve the performance. Hence, we only use a subset of the frames for training.

## A.3 CROSS-DATASET EVALUATION FOR SFT VS GRPO EXPERIMENTS

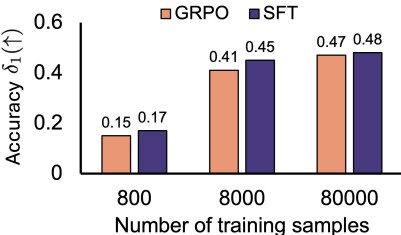

Figure 10: **SFT vs GRPO with cross dataset evaluation.** We show the result when we train on Argoverse2 and evaluate on NuScenes. The trend is similar as in Fig. 3b.

As mentioned in Sec. 3.2, SFT and GRPO have similar accuracy when trained on the same number of samples. Here, we show the same experiment with cross-dataset evaluation to verify whether GRPO can bring any benefit in the zero-shot scenario. We train the model on Argoverse2 and evaluate on NuScenes. As shown in Fig. 10, the trend of cross-dataset evaluation is similar to the evaluation on the same dataset.

## A.4 GRPO HYPER-PARAMETERS

### Table 5: **Hyper-parameters of different experiments.**

| Experiment | 3b SFT | 7b SFT | 12b SFT | 3b GRPO |
|---|---|---|---|---|
| Number of training samples | 35M | 60M | 23M | - |
| Learning rate | 5.2e-5 | 1.0e-4 | 5.6e-5 | 1.0e-5 |
| Learning rate schedule | Cosine with linear warmup | Cosine with linear warmup | Cosine with linear warmup | Constant |
| Warmup ratio | 0.1 | 0.1 | 0.1 | - |
| Batch size | 1280 | 1280 | 384 | 32 |
| FSDP | Yes | Yes | Yes | No |
| Gradient clipping | 0.1 | 0.5 | 1.0 | 1.0 |
| Gradient checkpointing | Yes | Yes | Yes | Yes |
| Bfloat16 | Yes | Yes | Yes | Yes |
| Flash Attention 2 | Yes | Yes | Yes | Yes |
| Unified Focal Length | 1000 pixels | 1000 pixels | 750 pixels | 1000 pixels |

For GRPO experiments, we use the group size of 8 with format reward, and set $\beta = 0$ during training, where $\beta$ is the weight in the GRPO loss terms to control the similarity between the outputs of the trained and the original models. Empirically changing the group size and learning rate schedules do not affect the performance significantly, but having non-zero $\beta$ tends to make the model underfit the metric depth task, leading to much lower accuracy and long but unnecessary reasoning traces. We apply format reward in GRPO, which did not improve accuracy but can make the model follow the output template. Other standard hyper-parameters are shown in the last column of Table 5. To make SFT hyper-parameters as close as possible to GRPO, we reduce the batch size to 32 during the experiment of SFT vs GRPO and reduce the learning rate correspondingly following the square root scaling rule (Li et al., 2024).

## A.5 PROMPTS FOR CAMERA AMBIGUITY ANALYSIS

As mentioned in Sec. 3.3, we compare different approaches for handling camera ambiguity to study what the best approach is for VLMs. To add camera intrinsics information explicitly into the text prompt, we follow the similar format of Seed1.5-VL, and ask: "*Given this image of size (width = W, height = H), where the camera intrinsics are (fx = A, fy = B, cx = C, cy = D) and the images are without distortions, how many meters is this point away from the camera?*". To let the model predict camera intrinsics explicitly, we add in the answer of each sample the camera ray direction before the actual metric depth, so that the model can condition on the predicted intrinsics to decide the metric depth more accurately. Specifically, we set the answer to "*The point is around X degrees to the right/left, Y degrees above/below and Z meters away from the camera.*" We do not predict the camera intrinsics (fx, fy, cx, cy) directly since even adding the exact values of them into the input prompt do not help.

## A.6 HYPER-PARAMETERS FOR DEPTHLM TRAINING

See Table 5 for the hyper-parameters of the models trained by SFT. For the 12B model from (Agrawal et al., 2024), we set the unified focal length to 750 pixels due to memory limits.

## A.7 DEPTHLM MULTI-TASK PERFORMANCE ON EACH DATASET

We report in Table 6 the multi-task performance of DepthLM on individual datasets.

### Table 6: **MultiTask performance on individual datasets.**

| Method | Out | | | Out+In | In | | | | Avg $\delta_1(\uparrow)$ |
|---|---|---|---|---|---|---|---|---|---|
| | Argo | DDAD | Nuscenes | ETH3D | ScanNet++ | sunRGBD | ibims1 | NYUv2 | |
| DISTANCE | 0.809 | 0.728 | 0.850 | 0.708 | 0.856 | 0.843 | 0.950 | 0.876 | 0.828 |
| PRINCIPAL AXIS DISTANCE | 0.809 | 0.683 | 0.850 | 0.744 | 0.846 | 0.856 | 0.960 | 0.896 | 0.831 |
| SPEED | 0.801 | 0.722 | 0.843 | 0.721 | 0.844 | 0.819 | 0.935 | 0.854 | 0.817 |
| TIME | 0.799 | 0.722 | 0.843 | 0.722 | 0.842 | 0.817 | 0.936 | 0.846 | 0.816 |
| TWO POINT DISTANCE | 0.651 | 0.423 | 0.670 | 0.479 | 0.784 | 0.727 | 0.760 | 0.759 | 0.657 |
| POSE | 0.989 | - | 0.762 | - | - | - | - | - | 0.876 |

