# OpenReview forum: "DepthLM: Metric Depth from Vision Language Models"
_ICLR.cc/2026/Conference — ICLR 2026 Oral_

### Official Review · Reviewer_w53k · 2025-10-23

**Soundness:** 4
**Presentation:** 4
**Contribution:** 4
**Rating:** 10
**Confidence:** 5

**Summary:**

This paper introduces DepthLM, which enables Vision-Language Model (VLM) to perform metric depth estimation with accuracy comparable to specialized vision models. The authors identify that VLM’s weakness in depth estimation is due to issues in pixel reference and camera ambiguity. To address this, DepthLM employs visual prompting for pixel localization and intrinsic-conditioned augmentation to unify focal lengths across datasets. Using only sparse text-based supervision (1 or 2-4 labeled pixels per image in large-scale training ), DepthLM outperforms large proprietary models like GPT-5 and Gemini-2.5-Pro and reaches parity with state-of-the-art pure vision models such as UniDepth.

**Strengths:**

1. DepthLM achieves expert-level metric depth estimation without changing the architecture or using complex losses in mainstream VLM. It’s very astonishing that VLM can beat expert depth estimation models in monocular depth estimation.
2. It’s also very astonishing that by querying a few pixels in each image, and tuning VLM to estimate depth for a few pixels of each image, VLM can beat expert depth estimation models. This finding is very interesting and meaningful.
3. The DepthLM framework extends beyond monocular depth estimation to other 3D reasoning tasks, like spatial VQA or robot manipulation, using one unified vision-language model. I think if general VLM can be tuned to be very good at depth estimation, it can also benefit downstream application that requires precise distance or spatial measurement, like robot navigation and manipulation.

**Weaknesses:**

**Major:**

1. The method requires significant computational resources. As stated in the paper, the training requires 23M-60M samples, with 2-4 pixels sampled per image, using 128 H100 GPU for 2-4 days. Nevertheless, I think if the training process can be integrated into the instruction-tuning stage of general VLM to improve depth perception ability, the computational overhead is acceptable.

**Minor:**

Since DepthLM explores leveraging VLM for depth estimation, the authors might consider discussing or citing a few related papers that also investigate VLM for depth estimation:
1. Zhang, R., Zeng, Z., Guo, Z., & Li, Y. (2022, October). Can language understand depth?. In Proceedings of the 30th ACM International Conference on Multimedia (pp. 6868-6874).
2. Hu, X., Zhang, C., Zhang, Y., Hai, B., Yu, K., & He, Z. (2024). Learning to adapt clip for few-shot monocular depth estimation. In Proceedings of the IEEE/CVF Winter Conference on Applications of Computer Vision (pp. 5594-5603).
3. Chen, W., Shi, C., Ma, C., Li, W., & Dong, S. (2024). DepthBLIP-2: Leveraging Language to Guide BLIP-2 in Understanding Depth Information. In Proceedings of the Asian Conference on Computer Vision (pp. 2939-2953).
4. Zeng, Z., Wang, D., Yang, F., Park, H., Soatto, S., Lao, D., & Wong, A. (2024). Wordepth: Variational language prior for monocular depth estimation. In Proceedings of the IEEE/CVF Conference on Computer Vision and Pattern Recognition (pp. 9708-9719).
5. Cui, B., Huang, Y., Bai, L., & Ren, H. (2025). TR2M: Transferring Monocular Relative Depth to Metric Depth with Language Descriptions and Scale-Oriented Contrast. arXiv preprint arXiv:2506.13387.
6. Zeng, Z., Wu, Y., Park, H., Wang, D., Yang, F., Soatto, S., ... & Wong, A. (2024). Rsa: Resolving scale ambiguities in monocular depth estimators through language descriptions. Advances in neural information processing systems, 37, 112684-112705.
7. Zhang, W., Liu, H., Li, B., He, J., Qi, Z., Wang, Y., ... & Jin, X. (2025). Hybrid-grained Feature Aggregation with Coarse-to-fine Language Guidance for Self-supervised Monocular Depth Estimation. In Proceedings of the IEEE/CVF International Conference on Computer Vision (pp. 6678-6692).
8. Zeng, Z., Ni, J., Wang, D., Rim, P., Chung, Y., Yang, F., ... & Wong, A. (2024). PriorDiffusion: Leverage Language Prior in Diffusion Models for Monocular Depth Estimation. arXiv preprint arXiv:2411.16750.

**Questions:**

1. In training, DepthLM conducts image resizing to unify focal length. Does inference also require unifying focal lengths, which requires camera intrinsics? If I understand correctly, unifying focal length during training is resizing all images to the same size, then by conducting random cropping, the model can generalize to different image sizes during inference. My question is, since the focal length is not unified during inference, could authors elaborate more about how the model can generalize to different image sizes with different focal lengths during inference?

2. This paper mentions that “Image diversity is more important than label density”. The experiment provides only an increase in the label density after the training samples (I assume it means the number of images) larger than 16M. I am quite curious about the contribution of label density and image diversity to the model’s performance, respectively. Could the authors provide the results that show the performance under the combination of different image diversity and label density? For example, draw several lines for Figure 4 (c), each line means sampling different numbers of pixels per training image. I know this experiment would require extensive computational resources, so it’s totally OK if only part of such an experiment could be shown.

3. How are the pixels sampled during evaluation? How many pixels are sampled per image, and how are they sampled? I didn’t find it in the paper.

4. How are labelled pixels sampled in each image? (How) Will the sampling strategy affect the performance in training?

---

> ### Author Response · Authors · 2025-11-21
> **Thanks for the positive comments and constructive feedbacks! See individual responses below.**
>
> **Computational resource:** The training cost of DepthLM is arguably reasonable. Though we can use more compute to maximize the accuracy, Fig.4 (c) of the main paper shows that 97.6\% of our performance can be obtained with 1/5 of the compute (7M samples), which corresponds to roughly 8 H100 + 6 days and provides a good trade-off for resource-limited scenarios. We will clarify this in camera ready.
>
> **Related work:** We will discuss these works in camera ready.
>
> **Unifying focal length:** In our default setting, yes we require the focal length information from camera intrinsics during both training and evaluation to unify the focal length. Moreover, unifying focal length is not resizing images to the same size, but to the same focal length following the equation in L291 of the paper. Actually after focal length unification, the images have varied sizes already during inference. In practice, for images that do not have ground-truth focal length, we can estimate it with reasonable accuracy leveraging recent camera intrinsics estimation methods such as "AnyCalib:
> On-Manifold Learning for Model-Agnostic Single-View Camera Calibration", or simply using the intrinsic estimation module of DepthPro or UnidepthV2. We will clarify this in camera ready.
>
> **Number of images vs number of labels:** We have included in Fig. 10 of the revised paper the corresponding result. Specifically, we train the model with a fixed number of question-answer (QA) pairs but vary the number of labeled points used per image to create the QAs. The result shows that given the same training budget of dataset size, increasing the label density while decreasing the number of training images will reduce the model accuracy.
>
> **Number of sampled pixels during evaluation:** As mentioned in L138-139 of the main paper, 8192 sampled pixels are used in all evaluation datasets to provide stable statistics.
>
> **How to sample labeled pixels in each image:** For simplicity, we perform random sampling with equal weights for all labeled pixels in a single image, i.e., no balanced sampling is applied.

---

> > ### Comment · Reviewer_w53k · 2025-11-24
> >
> > Hi,
> >
> > Thank you very much for the reply. To clarify, "8192 sampled pixels are used in all evaluation datasets." Does that mean 8192 images are used, with 1 pixel randomly sampled within each image?
> >
> > Other than that, I think all of my concerns have been addressed properly, and I would like to keep my original rating.

---

> ### Author Response · Authors · 2025-11-24
>
> Yes, if the dataset has >8192 images, then we will sample 8192 images, and each image will have 1 pixel sampled for evaluation. Otherwise, we will sample 8192 pixels with some sharing the same image.
>
> We would like to thank the reviewer for the response, and happy that we can address all the concerns!

---

### Official Review · Reviewer_r3mG · 2025-10-31

**Soundness:** 2
**Presentation:** 3
**Contribution:** 2
**Rating:** 4
**Confidence:** 4

**Summary:**

The goal of this paper is to investigate whether it is possible to fine-tune a pre-trained VLM to achieve high accuracy in per-pixel metric depth estimation without modifications to the model architecture or training objective.

The authors fix the model architecture and conduct a set of experiments to investigate the best way to incorporate task-specific information for 3D depth estimation into the model's input prompt to achieve high-accuracy depth predictions. They fine-tune VLM to perform the 3D depth estimation task by generating a target distance in text modality.


The contributions of this work are as follows:
1) an approach for training VLMs to achieve high accuracy in per-pixel depth estimation
2) an accompanying benchmark dataset (mix of public datasets) adapted to the training and evaluation of VLMs on the 3D depth estimation task.

**Strengths:**

1) Approach works with two different VLM architectures.
2) It is the first VLM-fine-tuning-based approach for metric depth estimation that reaches specialised pure vision models.

**Weaknesses:**

1) It is not clear how the proposed pretraining unlocks 3D understanding. It looks like authors use the terms "3D understanding" and "per-pixel 3D metric depth estimation" interchangeably. 3D understanding is much broader than per-pixel 3D metric depth estimation. See questions.
2) High training cost. The model is trained on ""128 H100 GPUs for about 2-4 days"" compared to 6 days on 16 NVIDIA 4090 with half precision (best monocular metric depth estimation baseline UniDepthV2).

**Questions:**

1) Fine-tuned model does achieve the result comparable to specialized pure vision models, but it is fine-tuned just on the per-pixel depth estimation task. What is the difference vs expert pure vision approaches, except that the model is trained to output the text? Even in "Beyond metric depth" experiments, when the model is trained on five other tasks (which in fact are all depth estimation related), what original VLM properties, except executing these 5 tasks on which it was trained, are preserved?

2) Related to the above question: L37-39: Is it still a VLM if it is trained to execute one or five specific tasks?

3) L117-118: how numerical 3D labels are chosen?

4) L324: "We tune the prompt for VLMs that are not trained directly on our task to maximize their performance". How was the prompt tuning done? Were all VLMs, selected as baselines, benchmarked using visual prompting?

5) L420: It -> it.

---

> ### Author Response · Authors · 2025-11-21
> **Thanks for the positive comments about the generality and performance of our method. We address individual concerns below and will improve the paper according to the constructive feedbacks**
>
> **3D understanding:** As mentioned in the abstract and intro of the paper, though "3D understanding" covers a broader range of tasks, "per-pixel metric depth estimation" is a widely used one to verify that a model understand the 3D scene. As shown in the multi-task part of the paper, the same framework can be extended to cover other 3D understanding tasks (e.g., camera pose estimation, multi-point relations) once the key problem of enabling the metric depth estimation is resolved. Therefore, we use "3D understanding" to discuss the broader picture of using VLMs for flexible 3D understanding tasks, but use "per-pixel metric depth estimation" to discuss specific points on enabling our representative task of metric depth estimation. We are happy to replace 3D understanding with other terms if the reviewer has better suggestions. We will also add a small discussion paragraph in camera ready to further clarify our scope.
>
> **Training cost:** The training cost of DepthLM is arguably reasonable. Though we can use more compute to maximize the accuracy, Fig.4 (c) of the main paper shows that 97.6\% of our performance can be obtained with 1/5 of the compute (7M samples), which corresponds to roughly 8 H100 + 6 days. Such compute budget is arguably acceptable comparing to 16 NVIDIA 4090 + 6 days, given the potential of DepthLM to provide standard VLMs with strong and flexible 3D understanding capabilities. We will clarify this point in camera ready.
>
> **Difference with expert pure vision model:** The major difference is the flexibility and unified architecture for diverse 3D tasks. We proved that DepthLM can enable specialist level 3D understanding capabilities in standard VLMs, it is straight-forward to mix DepthLM training data with general visual instruct tuning datasets during the instruct tuning phase of a general VLM to achieve high accuracy on both existing VLM tasks (i.e., preserving all original VLM properties) and 3D understanding tasks such as depth estimation and the tasks demonstrated in the "beyond metric depth" section. Previous papers have already proved the effectiveness of such strategy, e.g., ``SpatialRGPT: Grounded Spatial Reasoning in Vision-Language Models", where they show that mixing general instruct tuning datasets with 3D understanding data can enable new tasks while preserving the performance on conventional VLM datasets such as MMMU, SEED etc.. To show that this strategy works as expected, we are currently working on a similar experiment, which takes some time and resources to setup. We will update the paper and add follow-up comments if we can finish it before the rebuttal deadline, if not, we will provide the result in camera ready.
>
> **L37-39:** The architecture is fully preserved, and as mentioned above, we can do mix-data training to preserve the original VLM capabilities.
>
> **L117-118:** We assume the question is about how we sample sparsely labeled pixels to train our model. It is simple random sampling from all labeled pixels in a single image. Specifically, we will first randomly sample a batch of images during each training step, and for each sampled image, we randomly sample 1 pixel out of all labeled pixels with uniform weights, i.e., without any complex sampling strategy such as balanced sampling.
>
> **L324:** The prompt tuning was done by trying out 1) prompts that are closer to the suggested formats in the github repo of the released models, e.g., for Seed1.5-VL, we tried both thinking and non-think models with the official prompts provided in their github repo and pick the best result as the Seed1.5-VL (Official Setup) 2) the prompts modified from DepthLM prompts, with both text-based and visual-prompting-based prompts in the candidates. We will clarify this in camera ready.
>
> **We hope the above responses have addressed the major concern about 1) the 3D understanding terminology and 2) computational cost. And we look forward to have further discussions with the reviewer so that we can fully address all raised issues.** Given that both the performance, and the innovation on "being the first work to enable expert level 3D understanding capabilities on standard VLMs" have been consistently agreed by all reviewers, we kindly asks the reviewer to consider raising the score in the final review, thanks!

---

### Official Review · Reviewer_5nYc · 2025-11-01

**Soundness:** 3
**Presentation:** 3
**Contribution:** 3
**Rating:** 6
**Confidence:** 5

**Summary:**

This paper introduces DepthLM, a vision-language model designed for single-image depth estimation. The authors explore four key aspects: (1) how to prompt a VLM with a query point, (2) whether SFT or RL is more effective, (3) how to handle focal-length generalization, and (4) how much supervision per image is required. The method achieves promising results across experiments.

**Strengths:**

Clear and focused presentation. The paper is well-structured and easy to follow. Each experimental question is clearly motivated, investigated, and summarized. The writing is concise and avoids unnecessary technical ornamentation, which I find appealing.

Insightful experimental scope. The paper systematically studies several important dimensions of this task (prompting strategy, training scheme, focal length handling, and supervision density), offering valuable insights for the community.

Strong empirical performance. The model achieves competitive or superior results compared to prior methods, demonstrating practical effectiveness.

**Weaknesses:**

Overemphasis on visual prompting novelty. Prompting in the visual space instead of textual space is useful, but not highly innovative by itself. The paper seems to overstate the novelty of this contribution, dedicating perhaps too much discussion to it relative to its conceptual depth.

SFT vs. RL conclusion lacks depth. The conclusion that SFT outperforms RL feels premature. RL performance can vary significantly with algorithm choices (e.g., GRPO, PPO, DPO), reward design (e.g., continuous metric rewards vs. token-based or discrete depth-bin rewards), and training strategies (e.g., RL after SFT rather than combined in one stage). A more thorough investigation—or at least acknowledgment of these factors—would strengthen the claim.

Focal-length generalization clarity. The discussion on focal-length generalization is somewhat unclear. It appears the model is trained with unified focal-length settings and expected to generalize to unseen focal lengths during inference. Clarifying the exact assumption and mechanism would be helpful.

**Questions:**

see weakness

---

> ### Author Response · Authors · 2025-11-21
> **Thanks for your positive comments and constructive feedbacks. Individual responses are shown below.**
>
> **Visual prompting:** We are happy to adjust the statements according to the reviewer's suggestion, and agree that our discussion about visual prompting is not meant to emphasize its complexity but rather the the surprising practical effectiveness compared to text-based pixel reference, which makes it an important method component.
>
> **SFT vs RL:** We focus on comparing SFT with GRPO which was arguably the most widely used RL method at the time to train LLMs. We agree that this experiment did not cover all possibilities in RL since SFT works already reasonably well with high efficiency. We will discuss this in camera-ready to acknowledge the potential factors according to the reviewer's suggestion.
>
> **Focal-length:** We will clarify the setup in the implementation part. Specifically, the setup is similar to Metric3D, where we unify the focal length using camera intrinsics during both training and evaluation. For practical data that do not have attached focal length information, one obtain a reasonable estimate using existing single image intrinsic estimation methods such as "AnyCalib:
> On-Manifold Learning for Model-Agnostic Single-View Camera Calibration", or simply using the intrinsic estimation module of DepthPro or UnidepthV2.
>
> We look forward to have further discussions with the reviewer so that all the concerns can be fully addressed.

---

### Meta-Review · Area_Chair_BpgA · 2026-01-05

**Summary:**

The paper proposes DepthLM, a method to enable standard Vision Language Models (VLMs) to perform per-pixel metric depth estimation. By utilizing visual prompting, intrinsic-conditioned augmentation, and sparse text-based supervision, the authors demonstrate that VLMs can achieve accuracy comparable to specialized pure vision models without architectural modifications or complex loss functions.

The submission received a split range of scores (10, 6, 4), but Reviewer w53k (10) presents a compelling argument for the significance of the results, specifically, the surprising finding that sparse text supervision allows generalist VLMs to match specialist vision models.

While Reviewer r3mG's concerns about compute resources are valid, they are characteristic of foundation model research and the authors provided evidence that 97% of performance is recoverable with significantly less compute. The methodology opens a new pathway for 3D tasks in VLMs.

**Reviewer Concerns:**

Reviewer Concerns Addressed: The authors successfully clarified the focal length unification process (requiring intrinsics) and the sampling strategy for sparse labels. They also agreed to temper claims regarding the novelty of visual prompting and promised to include missing related works suggested by Reviewer w53k.

Outstanding: Reviewer r3mG retained reservations regarding the high computational cost (128 H100 GPUs), despite the authors presenting a lower-compute trade-off. Additionally, the debate on whether "3D understanding" is the appropriate terminology for depth estimation alone remains a point of semantic contention for r3mG, though the authors argued it is a representative task.

**Reviewer Scores:**

Reviewer w53k (10) and Reviewer 5nYc (6) has maintained high scores from the beginning.

Reviewer r3mG (4): This reviewer might have moved to a 5 or 6 if they were more convinced by the "mix-data" argument regarding the preservation of original VLM capabilities and if they viewed the computational cost as acceptable for a foundation model approach.

---

### Decision · Program_Chairs · 2026-01-26

Accept (Oral)